Comparison between two- and three-dimensional scoring of zebrafish response to psychoactive drugs: identifying when three-dimensional analysis is needed

Macrì Simone 1 2
http://orcid.org/0000-0003-1970-8386 Clément Romain J.G. 1
Spinello Chiara 1
http://orcid.org/0000-0002-1480-3539 Porfiri Maurizio 1 3 mporfiri@nyu.edu
1 Department of Mechanical and Aerospace Engineering, New York University, Tandon School of Engineering , Brooklyn, NY , USA
2 Centre for Behavioural Sciences and Mental Health, Istituto Superiore di Sanità , Rome , Italy
3 Department of Biomedical Engineering, New York University, Tandon School of Engineering , Brooklyn, NY , USA
Esteban María Ángeles
Electronic publication date: 2019 Oct 16
Publication date: 2019
Volume: 7
Electronic Location ID: e7893
Received 2019 Aug 1; Accepted 2019 Sep 16
Copyright: © 2019 Macrì et al.
Copyright year: 2019
Copyright holder: Macrì et al.
License: This is an open access article distributed under the terms of the Creative Commons Attribution License, which permits unrestricted use, distribution, reproduction and adaptation in any medium and for any purpose provided that it is properly attributed. For attribution, the original author(s), title, publication source (PeerJ) and either DOI or URL of the article must be cited.
License URL: https://creativecommons.org/licenses/by/4.0/

Keywords: Anxiety, Automated tracking, Citalopram, Ethanol, Novel tank diving test

Funding: National Institutes of Health, National Institute on Drug Abuse 1R21DA042558-01A1 Behavioral and Social Sciences Research that co-founded the National Institute on Drug Abuse National Science Foundation CMMI-1505832 This work was supported by the National Institutes of Health, National Institute on Drug Abuse under grant number 1R21DA042558-01A1, the Office of Behavioral and Social Sciences Research that co-funded the National Institute on Drug Abuse grant, and by the National Science Foundation under grant number CMMI-1505832. The funders had no role in study design, data collection and analysis, decision to publish, or preparation of the manuscript.

==============================
Zebrafish (Danio rerio) have recently emerged as a valuable laboratory species in the field of behavioral pharmacology, where they afford rapid and precise high-throughput drug screening. Although the behavioral repertoire of this species manifests along three-dimensional (3D), most of the efforts in behavioral pharmacology rely on two-dimensional (2D) projections acquired from a single overhead or front camera. We recently showed that, compared to a 3D scoring approach, 2D analyses could lead to inaccurate claims regarding individual and social behavior of drug-free experimental subjects. Here, we examined whether this conclusion extended to the field of behavioral pharmacology by phenotyping adult zebrafish, acutely exposed to citalopram (30, 50, and 100 mg/L) or ethanol (0.25%, 0.50%, and 1.00%), in the novel tank diving test over a 6-min experimental session. We observed that both compounds modulated the time course of general locomotion and anxiety-related profiles, the latter being represented by specific behaviors (erratic movements and freezing) and avoidance of anxiety-eliciting areas of the test tank (top half and distance from the side walls). We observed that 2D projections of 3D trajectories (ground truth data) may introduce a source of unwanted variation in zebrafish behavioral phenotyping. Predictably, both 2D views underestimate absolute levels of general locomotion. Additionally, while data obtained from a camera positioned on top of the experimental tank are similar to those obtained from a 3D reconstruction, 2D front view data yield false negative findings.

Introduction

Preclinical animal models constitute a central tool to detail the fundamental mechanisms underlying the expression of human emotions in physiological and pathological conditions (Haller & Alicki, 2012). Within this framework, several experimental models have been proposed to investigate the neurobiological processes underlying anxiety (Davis et al., 2010; Hart et al., 2010), an evolutionarily preserved adaptive emotion, normally occurring as an anticipatory response to a potential threat (Bateson, Brilot & Nettle, 2011). The adaptive value of anxiety resides in the fact that it limits the negative outcomes associated with a potential threat (Nesse, 1999). Notwithstanding its adaptive nature, inappropriate (context-independent) or excess anxiety may often culminate in anxiety-related disorders that require medical attention (Bateson, Brilot & Nettle, 2011).

In parallel with the aforementioned evolutionary roots, the underlying biological determinants of anxiety are very well conserved across different taxa. For example, the neuroendocrine machinery activated in response to external stressors exhibits striking homologies and analogies among fish (Bernier & Peter, 2001), birds (Lynn & Kern, 2018), rodents (Macrì & Wurbel, 2006), monkeys (Parker et al., 2012), and humans (Rodrigues, LeDoux & Sapolsky, 2009). Likewise, neurotransmitters such as serotonin have been associated with anxiety-related behaviors in species as diverse as fish (Fossat et al., 2014), birds (Hogg et al., 1994), humans (Caspi et al., 2003), and sheep (Lee et al., 2016).

Although rodents have traditionally constituted the species of choice in this field (Hart et al., 2010; Kalueff, Wheaton & Murphy, 2007), zebrafish have recently emerged as an extremely promising experimental species (Fontana et al., 2018; Shams et al., 2018; Stewart et al., 2014). The success of this freshwater species rests upon several advantages that range from genetic and neuroanatomic similarities between zebrafish and humans (Howe et al., 2013), to their small size and high reproductive rates favoring the execution of high-throughput studies (Kalueff, Stewart & Gerlai, 2014). In addition, the possibility to dissolve substances in water allows for the non-invasive administration of drugs readily absorbed through the gills (Tran & Gerlai, 2013). These characteristics designate zebrafish as a fundamental tool in the field of psychopharmacology, whereby they allow the preliminary screening of numerous drugs within spaces and time frames much smaller than those required by laboratory mammals (McCarroll et al., 2016).

High-throughput behavioral experiments on zebrafish generally share the following methodological structure: administration of experimental treatments (e.g., administration of water-soluble compounds and exposure to fear-evoking or social stimuli), videorecording of observable phenotypes, offline scoring of video, coding of the observed behaviors, and data analysis (Stewart et al., 2014). Traditional behavioral phenotyping leveraged the use of a single camera positioned on top or in front of the experimental tank and the subsequent use of behavioral scoring software, in which the phenotype of interest had to be input by a trained observer (Cianca et al., 2013; Spinello, Macrì & Porfiri, 2013). Albeit extremely productive, this approach was prone to observer bias and has been recently complemented by tracking algorithms capable of automatically coding and scoring zebrafish behavior with limited human supervision (Delcourt et al., 2018; Franco-Restrepo, Forero & Vargas, 2019; Nema et al., 2016; Perez-Escudero et al., 2014).

However, from the two-dimensional (2D) view offered by a single video-camera it is impossible to phenotype the three-dimensional (3D) swimming pattern exhibited by zebrafish. This consideration prompted the design and development of experimental platforms capable of investigating zebrafish behavior adopting a 3D approach (Cachat et al., 2011; Maaswinkel, Zhu & Weng, 2013; Macrì et al., 2017; Stewart et al., 2015). We recently demonstrated that the limitation of 2D scoring methods extends beyond the geometrical underestimation of swimming paths (3D trajectories being longer than their 2D projections by definition), and may result in numerous false positive and false negative findings (Macrì et al., 2017). Specifically, we first tested zebrafish in conventional binary choice behavioral assays, and then analyzed group differences based on 3D or 2D (top and front views) trajectories. This analysis demonstrated that 2D views generated approximately 20% of false findings, being represented by inappropriate reporting of significant inter-group differences in spite of indistinguishable ground truth data (false positives) or failure to detect significant results in instances in which experimental groups belonged to different populations (false negatives) (Macrì et al., 2017).

In the present study, we aimed at prospectively investigating whether 3D scoring of zebrafish behavior may also benefit zebrafish research in the field of behavioral response to psychoactive substances. To this aim, we exposed experimentally naïve zebrafish to drugs capable of modulating anxiety-related behaviors in both humans and zebrafish (Cianca et al., 2013; Sackerman et al., 2010), and then analyzed their phenotype in response to an anxiety-provoking test paradigm in 3D or in 2D (top and front views). Specifically, we investigated the behavior of zebrafish in a novel tank diving test in response to the administration of the selective serotonin reuptake inhibitor citalopram (30, 50, and 100 mg/L) or ethanol (0.25%, 0.50%, and 1.00%). These substances were selected based on their known efficacy in modulating anxiety-related behavior, their water-soluble nature, and their non-restricted use. These factors contribute to the replicability of the study and reproducibility of the results, while shedding light on the effects of widely used and abused substances (Pannia et al., 2014; Sackerman et al., 2010). The novel tank diving test has already been validated as a locomotion- and anxiety-related behavioral test (Cachat et al., 2010). Therein, anxiety is measured through the evaluation of fish position in the water column, swimming speed, erratic movements, and freezing, as functions of the time spent in the experimental tank from the initial release.

The objective of this study was twofold: first, we sought to replicate existing findings indicating that ethanol (Pannia et al., 2014) and citalopram (Sackerman et al., 2010) modulate anxiety in zebrafish (predictive validity of the assay), and then we aimed at testing whether the experimental advantages afforded by 3D scoring in drug-free states (Macrì et al., 2017) also extend to zebrafish psychopharmacology. Grounded in our previous work, we anticipated 2D views to be characterized by reduced absolute locomotion values compared to 3D trajectories. Most importantly, in the light of the high rate of false findings observed in drug-free conditions (Macrì et al., 2017), we expected the predictive validity of 2D trajectories to be potentially jeopardized. This hypothesis rests on the fact that, when exposed to psychoactive substances, fish may exhibit a series of responses that vary in space and time. For example, increased anxiety may reflect in a progressive reduction in general locomotion, increased freezing, erratic movements, and preference for the bottom of the experimental tank. These patterns manifest differentially depending on the time spent in the experimental apparatus (with preference for the bottom varying with the prolonged exposure), and on the view (i.e., top or front view). For example, while horizontal erratic movements are best detected through a top view, geotaxis can be appropriately scored only from a side view. Therefore, we hypothesized that the specific view may reflect into a bias in detecting time-dependent effects of psychoactive drugs, thereby potentially generating view × drug × experimental-progression effects.

Materials and Methods

Animal care and maintenance

The experiments and analyses were performed and reported according to the ARRIVE guidelines (Kilkenny et al., 2010). A total of 112 wild-type adult zebrafish (Danio rerio), with a 1:1 male/female ratio were used in this study. The fish were purchased from Carolina Biological Supply Co. (Burlington, NC, USA), and housed in 10 L (2.6 gallons) vivarium tanks (Pentair Aquatic Eco-Systems Locations, Cary, NC, USA), with a density of no more than 10 fish per tank. Fish were kept under a 12 h light/12 h dark photoperiod (Cahill, 1996), and fed with commercial flake food (Nutrafin max; Hagen Corp., Mansfield, MA, USA) once a day, approximately at 7 PM. Water parameters of the holding tanks were regularly checked, and temperature and pH were maintained at 26 °C and 7.2 pH, respectively. Regular tap water was used with the addition of a stress coat (AquaSafe plus, Tetra; Spectrum Brands Inc., Sulzbach, Germany) to remove chlorine and chloramines. Prior to the beginning of the experiments, fish were acclimatized in the holding facility for a period of 12–15 days.

The number of fish used in the study—compatible with obtaining sufficiently reliable and biologically relevant data—was estimated through a power analysis. Briefly, we computed the minimum required sample size considering the two-tailed Student t-test for independent groups using the following values, based on the results of previous studies (Abaid et al., 2012; Cianca et al., 2013; Spinello, Macrì & Porfiri, 2013): (i) standard deviation homogeneous among groups s = 0.23; (ii) type I error probability a = 0.05 and power 1 − b = 0.80 (conventional values); and (iii) minimum difference between control and treatment group means D = 0.17. The sample size resulting from this calculation was 15 subjects per group. To promote the generalizability of our findings, we conducted experiments on both males and females. We thus increased the sample size to 16 per group (eight males and eight females). We estimated that a sample size of 16 subjects (per group) would have 80% power to detect a 0.60 effect size on the principal outcome measures with a two-tailed significance level of 0.05.

Experimental setup

To obtain 3D trajectories, we used two Flea 3 USB high resolution cameras (Point Grey Research Inc., Richmond, BC, Canada), one overhead and one in front. Videos were acquired at 30 frames per second, using a resolution of 588 × 264 pixels from the top view and 652 × 360 pixels from the front view; the typical size of one video from the top view was six to seven gigabytes, while the front view was nine to 10 gigabytes. The dimensions of the test tank were 29 cm (length) × 14 cm (height) × 8.5 cm (width) and water 13 cm deep, similar to tanks used in comparable studies (Egan et al., 2009). The distance from the overhead camera and the bottom of the tank was 62 cm; while the distance from the front camera and the back of the tank was 60 cm.

To maximize the visual contrast and ease automatic tracking, the bottom of the tank was lined with white contact paper. The two short sides of the tank were covered with black contact paper to prevent reflection. On the other hand, the two long sides were kept transparent to allow data acquisition and avoid position bias (i.e., a potential side preference had one side been kept transparent for data acquisition and the other kept opaque). The experimental arena was surrounded by black curtains to prevent light reflection and visual disturbance from the outside.

Experimental procedure

Experiments, performed in June 2018, were conducted on seven groups, each consisting of 16 subjects (eight males and eight females). Specifically, the experimental design entailed one control group exposed to vehicle (water), three groups treated with citalopram (30, 50, 100 mg/L), and three groups treated with ethanol (0.25%, 0.50%, 1.00% ethanol/water solution in volume/volume %). The fish were randomly allocated to each of the seven conditions in the following way. The conditions were randomly distributed over several weeks, testing eight subjects per day (four in the morning and four in the afternoon). We balanced sex across conditions, and conditions across mornings and afternoons. Male and female fish were kept in separate tanks; in total, fish were housed in 12 tanks. At the beginning of each test session, we sampled one subject from a tank. Such a tank was different from that out of which we chose the previous subject tested in the same condition. This procedure guaranteed that potential tank effects were distributed evenly across all experimental groups.

Due to technical issues, four trials had to be discarded: this resulted in a slight reduction in the number of subjects in the 100 mg/L citalopram group (15 subjects instead of 16) and in both the 0.25% and 1.00% ethanol groups (14 and 15 subjects instead of 16, respectively). Based upon Sackerman et al. (2010), we measured the effect of acute exposure to citalopram by treating the fish to the substance for 5 min before testing it. Following previous work on the effect of ethanol by our group (Cianca et al., 2013), we measured the effect of exposure to ethanol over a 1-h period. In the interest of reducing the number of subjects used in animal experimentation, the same control subjects were used to test the effects of citalopram and ethanol. Fish were treated and tested in isolation.

Since these substances required a differential pre-exposure time (5 min for citalopram and 1 h for ethanol), we devised a common procedure for vehicle, citalopram, and ethanol. Thus, 1 h before testing, fish were placed in a 500 mL beaker filled with 450 mL of the following fluid: water for control and citalopram groups, or a solution of ethanol (0.25%, 0.50%, and 1.00%) for the other groups. 5 min before testing, an additional 50 mL of fluid were slowly added to the beaker over a period of 20–30 s. These 50 mL were constituted by either water (the control group), an ethanol solution of the concentration already present in the beaker (the ethanol groups), or a concentrated solution of citalopram that resulted in a final concentration in the beaker of 30, 50, or 100 mg/L (the citalopram condition). Fish were left in the beaker for 5 min, at the end of which they were transferred to the test tank and recorded for 6 min.

Simultaneous recording from both cameras was initiated before transferring the fish into the test tank. In addition, at the beginning of the recording, a laser beam, visible from both cameras, was pointed into the test tank in order to ensure later synchronization of both video streams. At the end of the experiment, the fish was hand-netted into a separate tank.

All the experiments were performed at the New York University Tandon School of Engineering (Brooklyn, NY, USA) in accordance with relevant guidelines and regulations, with National Institutes of Health guide for the care and use of Laboratory animals (NIH Publications No. 8023, revised 1978), and was approved by the University Animal Welfare Committee of New York University under protocol number 13-1424.

Tracking and 3D reconstruction

Images recorded from the high-resolution cameras were processed through an in-house developed tracking software, see Butail, Bartolini & Porfiri (2013) for a detailed description (the software is available for download at https://github.com/sach1tb/peregrine). The top and front view cameras provided time series of the trajectory projected onto the x-y and x-z planes, respectively. Each pair of tracks were automatically synchronized using the common x coordinate along length of the tank. The time-series for each x coordinate of the pair were shifted relative to each other and the relative shift producing the smallest difference was selected. Once synchronized, the tracks from the top view and from the front views were combined to construct the x, y, and z coordinates of the trajectory in the three-dimensional space (see Fig. 1 for a representative trajectory exhibited by a control subject).

Figure 1 Trajectory for a single fish from a control trial.

(A) Top view, (B) 3D reconstructed trajectory obtained from synchronizing trajectories from top and front views, and (C) front view. The color of the trajectory denotes the evolution of the position of the fish along the 6-min trial. The axes dimensions are 29 × 8.5 × 13 cm. Our an in-house developed tracking software (available for download at https://github.com/sach1tb/peregrine) was used in the analysis.

Reconstructed trajectories were used to quantify the following ethogram: time spent freezing (percentage of time that the fish moved less than two cm anywhere in the tank over a rolling period of 2 s), time spent wall following (percentage of time that the fish spent within three cm of any side wall or the bottom of the tank), average speed (time-average of the first-order numerical differentiation of the position time series), average peak speed (time-average of the speed values greater than the 90th percentile), average acceleration (time-average of the magnitude of the first order numerical differentiation of the velocity time series), average peak acceleration (time-average of the acceleration values greater than the 90th percentile), average angular speed (time-computed on the basis of a finite difference approximation of the curvature of fish trajectories), average peak angular speed (time-average of the angular velocity values greater than the 90th percentile), and time spent in the top half of the water column. These nine measures were selected from the technical literature on zebrafish behavior in novel tank tests (Cachat et al., 2010) and their objective scoring from 3D trajectories follows our previous work (Macrì et al., 2017; Mwaffo et al., 2015).

Statistical analyses

Experiments with ethanol and citalopram were analyzed separately, but both were compared to the same control condition.

Principal component analysis on 3D data

In order to detail the specific information that could be inferred from these measurements, we preliminarily conducted a principal component analysis (PCA) on nine behavioral measures, objectively scored from 3D trajectories (i.e., average speed, average peak speed, average angular speed, average peak angular speed, average acceleration, average peak acceleration, time spent freezing, time spent in the top half of the tank, and time spent in the vicinity of the walls). The PCA was aimed at detecting potential correlations among the variables and identifying underlying orthogonal factors associated with independent domains.

Only principal components with eigenvalues larger than one were retained in the analysis. For each compound (citalopram or ethanol), the loadings were varimax-rotated, and the resulting scores for each principal component were used as dependent variables in a four (citalopram: vehicle, 30, 50, and 100 mg/L, or ethanol: vehicle, 0.25%, 0.50%, and 1.00%) × six (time bins, 1 min each) × two (sex: male, female) repeated measures analysis of variance (ANOVA) for split-plot designs. Testing males and females served the aim to access a heterogeneous experimental population and therefore improve the generalizability of our findings.

For citalopram and ethanol treatments, three principal components with eigenvalue larger than one were extracted by the PCA (Table 1), accounting for 87% of the total variance. The first principal component, accounting for 47% of the variance, reflected locomotion, with positive loadings for average speed, average peak speed, average acceleration, and average peak acceleration, and a modest negative loading for the time spent freezing. The second principal component, accounting for 26% of the variance, reflected anxiety-related behavioral patterns (behavioral anxiety) with positive loadings for average angular speed, average peak angular speed, and the time spent freezing. The third principal component, accounting for 11% of variance, reflected anxiety-related spatial preference (positional anxiety), with positive loadings for the time spent wall following, and negative loadings for the time spent in top half.

Table 1 Principal component analysis.

Summary results from the principal component analysis for citalopram and ethanol conditions. Principal components with eigenvalue larger than 1 are shown. Loadings greater than 0.7 or smaller than −0.7 are emboldened; loadings smaller than 0.1 in magnitude are not displayed.

	Citalopram	Ethanol	
	Locomotion	Behavioral anxiety	Positional anxiety	Locomotion	Behavioral anxiety	Positional anxiety	
Eigenvalues	4.29	2.34	1.19	4.29	2.47	1.06	
Explained variance (%)	47.7	26.0	13.3	47.7	27.4	11.8	
Cumulative variance (%)	47.7	73.6	86.9	47.7	75.1	86.9	
Varimax-rotated loadings	
Speed	0.938	−0.261		0.924	−0.289		
Average peak speed	0.948	−0.172		0.953	−0.122		
Average angular speed		0.939			0.946	0.106	
Average peak angular speed		0.971			0.976	0.113	
Average acceleration	0.977			0.976			
Average peak acceleration	0.942			0.969			
Freezing	−0.542	0.760		−0.524	0.746	0.170	
Wall following		0.176	0.817		0.309	0.670	
Time in top half	−0.213		−0.788	−0.118		−0.873	

These three principal components, derived from 3D observations, were used to test the efficacy of ethanol and citalopram in modifying zebrafish behavior. It was not possible to use PCA to compare the different views since the number of variables that construct the principal components in 3D was greater than that in 2D.

Statistical model to compare 2D and 3D analyses

To investigate whether 2D projections of 3D trajectories may introduce a bias in the predictive validity of behavioral data on each of the nine measures, we conducted a repeated measures ANOVA for split-plot designs. In this analysis, the two general models for citalopram and ethanol were, respectively: three (view: 3D, 2D top, 2D front) × four (treatment: vehicle, 30, 50, 100 mg/L) × six (time bins, 1 min each) × two (sex: male, female), and three (view: 3D, 2D top, 2D front) × four (treatment: vehicle, 0.25%, 0.50%, 1.00% ethanol/water solution) × six (time bins, 1 min each) × two (sex: male, female) repeated measures ANOVAs. Similar to the PCA, predictions of the effect of sex were not considered.

For all ANOVAs, the distribution of the model residuals was visually inspected to verify that they were close to normality (Quinn & Keough, 2002). Statistical analyses were performed using R 3.5.0, with the aov function for ANOVAs, the prcomp function for the PCA, and the emmeans 1.3.0 package for post hoc comparisons using the Dunnett’s multiple comparisons test, comparing control to other conditions and first minute to other minutes.

While main effects of the view factor allowed assessing whether absolute values differed depending on the tracking method, significant interactions between view and any other factor suggested that the effects of the latter were moderated by the tracking method. For example, a significant view × treatment interaction would suggest that the effects of a given compound may vary as a function of how the behavior of the animal was scored (i.e., using 2D projections from top or front, or resorting to 3D trajectories). Upon detecting a significant interaction, we performed post hoc comparisons, correcting for type-I errors, to detail whether and which pairwise comparisons were significant. Among these comparisons, those contrasting 2D and 3D were germane to the key question of the study.

This statistical model allowed testing the hypothesis that 2D views yielded spurious results compared to 3D data: false positive or false negative findings. Specifically, if 2D analysis reported a significant inter-group difference which was not confirmed by 3D data, we recorded such a result as a false positive. Likewise, if 2D analysis failed to identify significant inter-group differences, which were instead detected using the 3D approach, we recorded such a result as a false negative. For example, consider the case in which we identified a significant time bins × view interaction where animals decreased their average speed as a function of time from 3D data while maintaining their initial average speed throughout the trial from both the 2D views. In this case, had post hoc comparisons between 3D data and any of the 2D data confirmed ANOVA results, we would register a false negative prediction for each of the 2D data. Instead, a false positive finding would be registered in the opposite situation where 3D data indicated the lack of a habituation profile in the average speed, while 2D data would support the opposite.

Results

Ethanol and citalopram alter individual habituation to the test, measured through PCA on 3D trajectories

When analyzing the three components identified by PCA, scored from 3D trajectories, we observed that absolute levels of locomotion were indistinguishable between control and citalopram-treated subjects (condition: F3,55 = 0.52, P = 0.668) (Fig. 2A). Additionally, general locomotion steadily declined throughout the experimental session in all subjects (time: F5,275 = 3.03, P = 0.011; t275 > 3.12, P < 0.009), regardless of the specific experimental group (time bins × condition: F15,275 = 1.13, P = 0.329). Absolute values of behavioral anxiety did not significantly vary across citalopram conditions (condition: F3,55 = 0.76, P = 0.524) (Fig. 2B). Yet, it significantly decreased over the trial (time: F5,275 = 3.52, P = 0.004; t275 > 2.69, P < 0.033), albeit at a different rate depending on the citalopram dose (time bins × condition: F15,275 = 1.85, P = 0.029). Specifically, while behavioral anxiety remained constant throughout the experimental session in citalopram 50 and 100 mg/L conditions, it significantly declined in control and citalopram 30 mg/L conditions (t275 > 2.60, P < 0.043). While positional anxiety did not significantly vary across citalopram conditions (condition: F3,55 = 0.95, P = 0.421) (Fig. 2C), it significantly increased over time (time: F5,275 = 3.51, P = 0.004; t275 > 2.82, P < 0.023). Such time-dependent profile varied depending on the experimental treatment (time bins × condition: F15,275 = 1.79, P = 0.036). Thus, while it remained constant in control and citalopram 100 mg/L, it was low at the beginning of the test session and steadily increased in citalopram 30 mg/L and citalopram 50 mg/L subjects (t275 > 2.74, P < 0.029).

Figure 2 Principal components for the citalopram conditions.

Mean ± standard error for (A) locomotion (B) behavioral anxiety, and (C) positional anxiety, over 6-min trials, showing overall variation, as well as for each concentration of citalopram (control 0, 30, 50, and 100 mg/L) based on the reconstructed trajectories in 3D. Data were analyzed through a repeated measures ANOVA for split-plot designs. Filled symbols denote a significant difference (P < 0.05) from the first minute within each condition. Horizontal bar denotes a significant overall difference in time.

In response to ethanol administration, absolute levels of locomotion failed to reach a statistically significant variation across experimental groups (condition: F3,52 = 2.43, P = 0.072) (Fig. 3A). When analyzing the time course of general locomotion, we observed that it significantly decreased over time (time: F5,260 = 3.02, P = 0.011; t260 > 2.67, P < 0.035), and that such a decrease was indistinguishable across all experimental groups (time bins × condition: F15,260 = 1.50, P = 0.105). Behavioral anxiety did not significantly vary across ethanol conditions (condition: F3,52 = 0.80, P = 0.500) (Fig. 3B), neither did it apparently change over time (time: F5,260 = 1.66, P = 0.144). However, we observed that the habituation profile varied depending on the specific experimental group (time bins × condition: F15,260 = 1.83, P = 0.031). Specifically, while behavioral anxiety remained constant in most experimental groups, it significantly declined over time in the ethanol 0.5% condition (P < 0.050; t260 = 2.89; P = 0.019). Finally, positional anxiety failed to reach a statistically significant variation across ethanol conditions (condition: F 3,52 = 2.49, P = 0.071) (Fig. 3C), although it significantly decreased over time (time: F 5,260 = 3.25, P = 0.007; t260 > 2.75; P < 0.029). Specifically, it significantly decreased for the ethanol 1.0% condition (time bins × condition: F15,260 = 2.33, P = 0.004; t260 > 3.76; P < 0.001).

Figure 3 Principal components for the ethanol conditions.

Mean ± standard error for (A) locomotion (B) behavioral anxiety, and (C) positional anxiety, over 6-min trials, showing overall variation, as well as for each concentration of ethanol (control 0%, 0.25%, 0.50%, and 1.0%) based on the reconstructed trajectories in 3D. Data were analyzed through a repeated measures ANOVA for split-plot designs. Filled symbols denote a significant difference (P < 0.05) from the first minute within each condition. Horizontal bar denotes a significant overall difference in time.

The scoring view influences the validity of experimental outcomes

Herein, we report data concerning the effects of the views on all the experimental variables measured in the study. For the sake of clarity, in this section, we only report statistical findings associated with the scoring view (3D, 2D top, and 2D front) and its interactions with time or condition. Results concerning the main effects of condition, time, and their interaction irrespective of view are available in the Supplemental Material.

Before delving into detailed comparisons between the three different views for all the considered behavioral measures, we present an aggregated assessment of potentially inaccurate conclusions that would be drawn from 2D projections against 3D trajectories. The rate of false negative (erroneous reporting of absence of differences in lieu of significant findings in 3D) and false positive (erroneous reporting of significant differences in lieu of non-significantly different findings in 3D) findings is synoptically reported in Tables 2 and 3.

Table 2 Number of false positive and false negative findings for citalopram.

Number of false positives and false negatives produced for each parameter when computed based on 2D top view and front view data, for the citalopram conditions. A false positive indicates that the 2D view (top or front) yields a significant result that is not supported by the 3D scoring approach. A false negative indicates that the 2D view (top or front) fails to detect a significant result that is instead evident from the 3D scoring approach.

Citalopram	Differences between 3D and 2D top view	Differences between 3D and 2D front view	
Parameters	False positives	False negatives	Total	False positives	False negatives	Total	
Average speed	1	0	1	0	0	0	
Average peak speed	0	0	0	0	2	2	
Average angular speed	0	0	0	0	3	3	
Average peak angular speed	1	0	1	0	2	2	
Average acceleration	0	0	0	0	3	3	
Average peak acceleration	0	0	0	0	2	2	
Wall following	0	0	0	0	2	2	
Time in top half	–	–	–	0	0	0	
Freezing	0	0	0	0	0	0	
Total	2	0	2	0	14	14	

Table 3 Number of false positive and false negative findings for ethanol.

Number of false positives and false negatives produced for each parameter when computed based on 2D top view and front view data, for the citalopram conditions. A false positive indicates that the 2D view (top or front) yields a significant result that is not supported by the 3D scoring approach. A false negative indicates that the 2D view (top or front) fails to detect a significant result that is instead evident from the 3D scoring approach.

Ethanol	Differences between 3D and 2D top view	Differences between 3D and 2D front view	
Parameters	False positives	False negatives	Total	False positives	False negatives	Total	
Average speed	0	0	0	0	0	0	
Average peak speed	0	0	0	0	1	1	
Average angular speed	1	0	1	0	1	1	
Average peak angular speed	1	0	1	0	1	1	
Average acceleration	0	0	0	3	0	3	
Average peak acceleration	0	0	0	0	0	0	
Wall following	0	0	0	3	0	3	
Time in top half	–	–	–	0	0	0	
Freezing	0	0	0	0	0	0	
Total	2	0	2	6	3	9	

Citalopram

Average speed: Predictably, average speed varied significantly depending on which view was used to compute it (view: F2,110 = 118.46, P < 0.001) (Fig. 4A). Specifically, both 2D front and top views underestimated absolute levels of locomotion compared to 3D data (t291.7 = 9.87, P < 0.001; and t291.7 = 6.88, P < 0.001, respectively); additionally, 2D front view resulted in reduced average speed compared to top view (t291.7 = 2.98, P = 0.009). Experimental subjects did not show a habituation profile to the test, yet 2D top projections indicated that the average speed decreased from the first to the last minute (time bins × view: F10,550 = 14.08, P < 0.001; t317.5 = 3.03, P = 0.012).

Figure 4 Behavioral parameters for the citalopram conditions.

Mean ± standard error for (A) average speed (B) average peak speed (C) average angular speed (D) average peak angular speed (E) average acceleration (F) average peak acceleration (G) proportion of time spent within three cm of walls (H) proportion of time spent in the top half of the tank, and (I) proportion of time spent freezing, over 6-min trials aggregated for all citalopram conditions, computed from 2D front and top views, and 3D reconstructed trajectories. Data were analyzed through a repeated measures ANOVA for split-plot designs. Filled symbols denote a significant difference (P < 0.05) from the first minute within each condition. Horizontal bar denotes a significant overall difference over time. Filled symbols in the top right corner of each panel indicate a significant overall difference with respect to 3D data.

Average peak speed: Average peak speed was significantly underestimated in both 2D front and top views in comparison with 3D data (view: F2,110 = 81.93, P < 0.001; t346.6 > 6.64, P < 0.001) (Fig. 4B). While the average peak speed decreased over time in all subjects (Supplemental Material), experimental groups showed a differential habituation profile (time bins × view: F10,550 = 15.64, P < 0.001; t286.4 > 2.73, P < 0.029).

Average angular speed: Average angular speed was underestimated in the 2D front view compared to both 3D and 2D top views (view: F2,110 = 88.45, P = 0.001; t295.3 > 5.33, P < 0.001) (Fig. 4C). Furthermore, a decrease in average angular speed over time was observed in all views (time bins × view: F10,550 = 7.40, P < 0.001; t275 > 2.69, P < 0.034).

Average peak angular speed: Average peak angular speed was underestimated when scored from 2D top view (view: F2,110 = 10.09, P < 0.001; t358.6 = 3.62, P = 0.001) (Fig. 4D). A decrease in average peak angular speed over time was recorded from all views, but not at the same times (time bins × view: F2,550 = 2.72, P = 0.003; t275 > 2.64, P < 0.038).

Average acceleration: Average acceleration was underestimated in both front and top 2D views compared to 3D (view: F2,110 = 126.62, P < 0.001; t284.4 = 9.24, P < 0.001; and t284.4 = 5.57, P < 0.001, respectively); additionally, 2D front view underestimated average acceleration compared to 2D top view (t284.4 = 3.67, P < 0.001). Average acceleration varied over time depending on the view adopted to score fish behavior (time bins × view: F10,550 = 13.40, P < 0.001) (Fig. 4E). Specifically, although average acceleration steadily declined from the third minute in ground truth 3D data (t371.0 = 2.57, P < 0.046), such a decline was observable also from 2D top view (t371.0 > 2.80, P < 0.025), but only during the last minute in 2D front view (t275.0 = 2.70, P < 0.033).

Average peak acceleration: Average peak acceleration significantly decreased over time, regardless of the specific view adopted to compute this measure (time bins × view: F10,550 = 4.42, P < 0.001; t344.7 > 3.15, P < 0.008) (Fig. 4F). Yet, average peak acceleration was underestimated in 2D front and top views compared to 3D (view: F2,110 = 79.67, P < 0.001; t402.3 = 6.05, P < 0.001; and t402.3 = 3.46, P = 0.002, respectively). Additionally, 2D front view yielded a lower average peak acceleration compared to top view (t402.3 = 2.59, P = 0.027).

Wall following: Time spent wall following was significantly underestimated in 2D front view compared to 3D data (view: F2,110 = 237.90, P < 0.001; t234.4 = 17.55; P < 0.001). Additionally, this metric was lower in 2D front view compared to 2D top view (t234.4 = 15.95, P < 0.001) (Fig. 4G). While 3D and 2D top view data indicated that wall following increased between the first and fifth minute of the experimental session (time bins × view: F10,550 = 2.16, P = 0.019, t824.7 > 2.56, P < 0.05), 2D front view data failed to identify this time dependent pattern of thigmotaxis.

Position in the water column (proportion of time spent in the top half): Since this metric takes into account only the vertical position of the fish, it cannot be scored from 2D top view and there is no difference between values from 2D front view and 3D reconstructed trajectories (Fig. 4H and the corresponding Fig. 5H for ethanol treatment).

Figure 5 Behavioral parameters for the ethanol conditions.

Mean ± standard error for (A) average speed (B) average peak speed (C) average angular speed (D) average peak angular speed (E) average acceleration (F) average peak acceleration (G) proportion of time spent within three cm of walls (H) proportion of time spent in the top half of the tank, and (I) proportion of time spent freezing, over 6-min trials aggregated for all ethanol conditions, computed from 2D front and top views, and 3D reconstructed trajectories. Data were analyzed through a repeated measures ANOVA for split-plot designs. Filled symbols denote a significant difference (P < 0.05) from the first minute within each condition. Horizontal bar denotes a significant overall difference over time. Filled symbols in the top right corner of each panel indicate a significant overall difference with respect to 3D data.

Freezing: Although the time spent freezing seemed to vary depending on which view was used to compute it (view: F2,110 = 4.19, P = 0.018) (Fig. 4I), post hoc tests revealed no pairwise difference. Similarly, although an interaction between view and time was registered (time bins × view: F10,550 = 2.35, P = 0.010), post hoc comparisons did not indicate any specific difference.

Ethanol

Average speed: The different scoring views resulted in variable average speed values (view: F2,104 = 90.45, P < 0.001) (Fig. 5A). Both 2D top and front views underestimated average speed compared to 3D (t174.3 = 3.79; P < 0.001; and t174.3 = 9.47; P < 0.001, respectively). Additionally, average speed was lower in 2D front view compared to top view (t174.3 = 5.69; P < 0.001). While data inspection suggested that habituation profiles were skewed by the view adopted to score individual behavior (time bins × view: F10,520 = 10.14, P < 0.001), post hoc analyses failed to show significant view-dependent variations in this parameter.

Average peak speed: Average peak speed varied in all subjects, and this profile was apparently influenced by the view adopted to score individual trajectories (time bins × view: F10,520 = 9.36, P < 0.001; t260.0 > 2.64, P < 0.039). This variation was manifested as a robust decline in subjects treated with ethanol 0.50% concentration (time bins × condition × view: F30,520 = 1.83, P = 0.005; t260.0 > 2.57, P < 0.047, see Fig. S2). Furthermore, average peak speed was significantly underestimated in both 2D front and top views compared to 3D data (view: F2,104 = 83.2, P < 0.001; t270.2 > 4.41; P < 0.001) (Fig. 5B), as well as from the front view compared to the top view (t260.0 = 2.60; P = 0.027). Although ANOVA reported a significant interaction between view and condition (condition × view: F6,104 = 2.33, P = 0.037), post hoc tests failed to reveal any significant pairwise difference.

Average angular speed: Predictably, both 2D front and top views underestimated average angular speed compared to the 3D view (view: F2,104 = 63.33, P < 0.001; t265.7 = 7.73, P < 0.001; and t265.7 = 4.78, P < 0.001, respectively). Additionally, average angular speed was smaller in 2D front view compared to 2D top view (t265.7 = 2.95, P < 0.010) (Fig. 5C). Furthermore, 3D data demonstrated that average angular speed declined between the first and the last minute of observation. Such a decline, observable in 2D top view data, was not detected in 2D front view (time bins × view: F10,520 = 5.19, P < 0.001, t730.9 = 3.902, P < 0.005).

Average peak angular speed: Although average peak angular speed appeared significantly different depending on which view was used (view: F2,104 = 4.37, P = 0.015), such a difference failed to emerge in post hoc comparisons. From all views, average peak angular speed declined throughout the experimental session (time bins × view: F10,520 = 3.53, P < 0.001; t707.1 > 2.66, P < 0.035) (Fig. 5D).

Average acceleration: In line with most of the locomotion-related variables, average acceleration was underestimated in both 2D front and top views compared to 3D view (view: F2,104 = 64.41, P < 0.001; t164.6 = 7.45, P < 0.001; and t164.6 = 3.20, P = 0.005, respectively, see Fig. 5E). Furthermore, 2D front view yielded lower values of the average acceleration compared to the top view (t164.6 = 4.25, P < 0.001). Individual habituation profile was differentially expressed by experimental subjects depending on the specific view (time bins × view: F10,520 = 11.87, P < 0.001). Specifically, while 3D and 2D top view data indicated a general decrease in average acceleration throughout the experimental session (t312.7 = 2.94, P < 0.016), such a profile was not visible in 2D front view, showing only a reduction during the third minute of the test (t260.0 = 2.70, P = 0.033). Data analysis suggested that the habituation profile varied depending on both the view and the ethanol treatment (time bins × condition × view: F30,520 = 1.50, P = 0.046). Specifically, we observed that the reduction in average acceleration was significant in ethanol 0.5% (t260.0 > 2.80, P < 0.025), and that this decrease occurred regardless of the specific view from which data were scored.

Average peak acceleration: Average peak acceleration varied depending on the specific scoring view (view: F2,104 = 61.85, P < 0.001); specifically it was underestimated in both 2D front and top views compared to 3D (t273.9 = 5.69, P < 0.001; and t273.9 = 2.41, P = 0.044, respectively) and was also less in 2D front view compared to 2D top view (t273.9 = 3.28, P = 0.003) (Fig. 5F). Furthermore, although data inspection suggested that the time-dependent habituation profile varied depending on the specific view (time bins × view: F10,520 = 5.70, P < 0.001; t312.6 = 2.57, P < 0.047), post hoc tests did not support this suggestion. Thus, acceleration decreased with time in experimental subjects regardless of the specific view adopted.

Wall following: The time spent in the proximity of the walls significantly varied depending on the specific view used to compute it (view: F2,104 = 56.09, P < 0.001). Wall following was significantly underestimated in both 2D front and top views compared to 3D (t178.8 = 9.96; P < 0.001; and t178.8 = 3.49; P = 0.002, respectively), and this parameter was less in 2D front view compared to 2D top (t178.8 = 6.47, P < 0.001). The individual habituation profiles varied depending on the view (time bins × view: F10,520 = 3.37, P = 0.001) (Fig. 5G). Specifically, wall following remained constant in 3D and 2D top view, and decreased in 2D front view (t260.0 = 2.79; P < 0.025). While wall following was apparently differed between conditions depending on the scoring view (condition × view: F2,104 = 5.54, P < 0.001), such a difference was not statistically significant in pairwise comparisons.

Freezing: While the time spent freezing seemed to vary depending on the specific scoring view (view: F2,104 = 5.35, P = 0.006) (Fig. 5I), such a difference was not confirmed by post hoc tests performed between the first and the sixth minute.

Discussion

The methodological nature of the present study first reverberated in the systematic evaluation of the correlation among the variables that constitute the ethogram exhibited in the novel tank diving test. The PCA revealed the presence of three orthogonal factors, reflecting general locomotion (average speed, average peak speed, average acceleration, and average peak acceleration), anxiety-related behavioral patterns (average angular speed, average angular peak speed, and freezing), and anxiety-related spatial preference (time spent close to the side walls and time spent in the top half of the water column). The first principal component relates to the translational motion within the water tank. The behavioral patterns loading on the second principal component have been consistently associated with anxiety, in the form of erratic movements (zig-zagging) and freezing (Kalueff et al., 2013). From the catalog of Kalueff et al. (2013), anxiety-related behavior is also related to thigmotaxis and geotaxis, which are the two behavioral measures that load on the third principal component.

While this analysis aligns with previous evidence indicating that anxiety can be expressed through different modalities, it also points at potential pitfalls of common practice in the construction of the ethogram of the novel tank diving test from 2D views. Specifically, the fact that variables contributing to the same principal component require different perspectives further corroborates the need for a 3D approach. For example, while position in the water column requires a front camera, wall distance and erratic movements need an overhead camera.

The analysis conducted on the aforementioned principal components revealed that both citalopram and ethanol influenced anxiety-related behaviors, thus corroborating the predictive validity of the novel tank diving test. Importantly, while citalopram concentration-dependently reduced locomotion and predictably reduced anxiety, ethanol resulted in increased anxiety, but only at a medium concentration (Tran et al., 2016b). Higher and lower ethanol concentrations were apparently ineffective. Low and medium concentrations of citalopram did not influence general locomotion but were associated with the exhibition of reduced anxiety, selectively during the first 3 minutes of testing. High concentrations of citalopram were associated with reduced locomotion and reduced anxiety throughout the entire test session. The anxiolytic effects of citalopram have already been reported in several studies. Sackerman et al. (2010) reported that zebrafish treated with 100 mg/L citalopram spent significantly more time than control fish in the top two thirds of the tank, suggesting a decrease in anxiety compared to the control.

It is worth noticing that, when analyzing discrete parameters rather than focusing on the principal components, some anxiety-related behavioral parameters seemed unaffected by the anxiolytic treatments applied. Specifically, we failed to observe a significant effect of citalopram on the time spent in the upper portion of the tank, a classical measure of anxiety. We note that such absence of a concentration-dependent behavioral response to anxiolytic compounds has also been reported in other studies. Sackerman et al. (2010) reported that acute exposure to 0.5% ethanol failed to alter the time spent in the upper portion of the test tank in zebrafish. Similarly, Maximino et al. (2011) failed to observe significant anxiety-related behavioral alterations in response to fluoxetine. Finally, in a previous study, we also observed that 0.25% and 0.5% ethanol did not modulate anxiety-related behaviors in the light-dark test (Cianca et al., 2013). These false negative findings further corroborate the potential heuristic value of conducting PCA in zebrafish behavioral pharmacology.

The anxiolytic effects of citalopram are likely related to its direct influence on serotonergic concentrations. For example, handling stress has been shown to increase anxiety-like behavior and reduce brain concentrations of the serotonin metabolite 5-Hydroxyindoleacetic acid (5-HIAA) (Tran et al., 2016a). Furthermore, Maximino et al. (2014) observed that acute administration of the 5-HT1a receptor agonist buspirone reduced behavioral anxiety in the light-dark test. Finally, in accordance with the present study, the acute administration of the selective serotonin reuptake inhibitor fluoxetine resulted in reduced anxiety in the geotaxis test (Maximino et al., 2013).

With respect to ethanol, available literature (Gerlai et al., 2000) indicates that its effects vary depending on the concentration, administration schedule, and methodological issues. Pannia et al. (2014) documented the emergence of numerous ethanol-induced behavioral changes, which are potentially influenced by zebrafish strain. Similar to our work, these changes often manifest in the form of a complex habituation profile, where the behavior of the animal varied as a function of time during the trial. Tran, Facciol & Gerlai (2016) reported that ethanol can have either anxiogenic or anxiolytic effects on zebrafish depending on whether the water in the test tank comes from the individual’s holding tank or from a tank that did not hold any fish.

Further, since ethanol influences general locomotion, some of its effects on anxiety may be spurious and potentially relate to locomotor effects. For example, a lack of vertical exploration may reflect a decrease in swimming behavior due to the sedative effect of high concentration of ethanol, rather than an anxiety response (Rosemberg et al., 2012). In our previous study (Cianca et al., 2013), we observed that high ethanol concentration resulted in reduced anxiety, associated with reduced motility and increased freezing. Likewise, Gebauer et al. (2011) observed that ethanol administration resulted in reduced anxiety in the light/dark test, but not in the novel tank diving test. In contrast with these findings, Tran et al. (2016b) reported that acute exposure to high ethanol concentration resulted in increased preference for the bottom of the test tank, and that such a variation related to alterations in brain monoamines. Specifically, alcohol-treated subjects showed reduced concentrations of the dopamine metabolite 3,4-Dihydroxyphenylacetic acid (DOPAC), of serotonin and its metabolite 5-HIAA (Tran et al., 2016b). Thus, while the effects of ethanol on anxiety are more variable compared to those exerted by citalopram, they apparently impinge on the same neurochemical pathways modulated by citalopram. Ultimately, the complementary use of these substances served the aim to address the validity of 2D approaches in zebrafish pharmacology of anxiety.

In order to compare 3D and 2D approaches, all experimental variables were also analyzed independently from one-another. This comparison was aimed at confirming the intuition that locomotion is underestimated when scoring the behavior in 2D and at assessing whether 2D views yielded incorrect conclusions regarding the effects of anxiolytics on individual behavior. Working with raw experimental variables rather than aggregated principal components allowed for a direct comparison of our findings with available literature, where the selected metrics are routinely assessed in pharmacological phenotyping of zebrafish (Kalueff et al., 2013). With respect to absolute values of locomotion, predictably, they were higher in 3D than 2D, regardless of whether the latter referred to the frontal or the horizontal plane. This can be easily explained by recognizing that 2D trajectories correspond to the projection of the full 3D motion on independent views, which would, by definition, abolish movement along a third dimension. This evidence echoes our previous findings obtained in drug-free states (Macrì et al., 2017).

The core objective of the present study was to evaluate whether 2D views may result in inaccurate rejection of null hypotheses or acceptance of alternative ones. We observed that the specific view consistently skewed the time course of the behavioral response to the novel tank. This was reflected in the presence of ubiquitous significant view × time bins interactions across most of the variables, and only few instances of view × condition interactions. Thus, these data could preliminarily suggest a relative robustness of current scoring methods in zebrafish pharmacology. Yet, in the light of the paucity of drug-dependent effects and of the nature of the statistical model required to test the suitability of the 2D approaches compared to 3D, we argue that this assessment only reflects a partial consideration of the observed results.

In our previous study, we demonstrated that 2D experiments are underpowered compared to 3D and therefore more prone to false negative findings than false positive ones (Macrì et al., 2017). While in situations characterized by few significant main effects of a given variable the likelihood to observe false negatives is intrinsically limited, data with numerous significant main effects shall be amenable to the identification of numerous false reporting instances. Accordingly, in the present study, the sporadic main effects of the condition have apparently masked view-dependent false negatives; complementarily, the ubiquitous presence of main effects of time bins allowed the detection of numerous view × time bins interactions. Thus, the specific view from which data were scored influenced the observed individual habituation patterns to the experimental paradigm. For example, while 3D data indicated that locomotion-related parameters (e.g., speed, angular speed, and acceleration) declined throughout the experimental session, 2D front view data failed to capture such a time-dependent habituation pattern. While this aspect may simply indicate the limited heuristic potential of the front view and advocate in favor of the use of a top view camera, we nonetheless note that a front camera is indispensable to quantify the position in the water column, which contributes to the anxiety-related phenotype.

These considerations extrapolate to zebrafish pharmacology, whereby our and others’ data (Cachat et al., 2010) indicate that anxiety-modulating compounds often alter habituation profiles rather than absolute values averaged across different time points (Wong et al., 2010). For example, we reported that anxiety-related behaviors in control subjects appear relatively constant throughout the entire course of the experimental session. Conversely, experimental subjects treated with low and medium concentrations of citalopram exhibit reduced anxiety-related behaviors during the early stages of the task, which gradually rise to attain control values toward the end of the session. Similar to Watts et al. (2017), we found that although 3D measures offer higher precision, the benefit of using 3D compared to a view from the top is limited regarding general behavioral pattern. The use of a front view remains necessary to capture specific behaviors linked to the position of the fish in the water column.

Conclusions

In the present study, we examined whether the effort required to analyze zebrafish behavioral response to psychoactive compounds in 3D is warranted or a simpler 2D approach should be preferred. Such an effort resides in the use of two, rather than one, cameras and in the increased computational burden required when dealing with multiple views. The former aspect adds some limited costs to the experiment and requires some design considerations to ensure that the experimental tank is visually accessible from multiple locations. The latter aspect entails an increased workload by the experimenter to manually repair tracks that the behavioral scoring software had flagged during the automated analysis, as well as limited computational costs associated with synchronizing and fusing multiple videos.

Although not conclusive, our data suggest that 2D analysis may produce spurious predictions that might confound the generalizability of experimental results. We acknowledge that our claims are specific to an experimental tank which allowed fish to freely swim along the water column. Testing animals in shallow water, restricting behavioral patterns to 2D, may not constitute an ideal solution, whereby it would add an unwanted source of distress to the animals that could mask the effect of the pharmacological manipulation. We also acknowledge that the experimental tank had a length threefold the width, which might have skewed the behavioral patterns of the animals with respect locomotion in the horizontal plane.

Finally, it is important to emphasize that in the present study we primarily focused on anxiolytic drugs and we thus cannot extrapolate our findings to the entire spectrum of anxiety-related behaviors. Future studies are needed to test whether the considerations outlined in this study also extend to anxiogenic compounds (e.g., caffeine) and non-pharmacological anxiety-eliciting stimuli (e.g., predators).

Supplemental Information

Supplemental Information 1 Supplementary material.

Click here for additional data file.

The authors are grateful to Hannah Kurdila for her help in data analysis, Shinnosuke Nakayama for his help with the statistics, and Boris Arbuzov for his assistance in scoring fish behavior from the video recordings.

Additional Information and Declarations

Competing Interests

Author Contributions

Animal Ethics

Data Availability

The authors declare that they have no competing interests.

Simone Macrì conceived and designed the experiments, analyzed the data, prepared figures and/or tables, authored or reviewed drafts of the paper, approved the final draft.

Romain J.G. Clément performed the experiments, analyzed the data, prepared figures and/or tables, authored or reviewed drafts of the paper, approved the final draft.

Chiara Spinello performed the experiments, authored or reviewed drafts of the paper, approved the final draft.

Maurizio Porfiri conceived and designed the experiments, analyzed the data, authored or reviewed drafts of the paper, approved the final draft.

The following information was supplied relating to ethical approvals (i.e., approving body and any reference numbers):

The NYU Washington Square Campus University Animal Welfare Committee (UAWC) provided full approval for this research (IACUC protocol #:13-1424).

The following information was supplied regarding data availability:

Data and code are publicly available on GitHub at https://github.com/DynSysLab/zebrafish_drug_3d.

Our in-house developed tracking software is available for download, also on GitHub: https://github.com/sach1tb/peregrine.

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
