# Peer review of "Comparison between two- and three-dimensional scoring of zebrafish response to psychoactive drugs: identifying when three-dimensional analysis is needed"

_PeerJ, doi:10.7717/peerj.7893_

## Round 0.1 · original submission · Major Revisions

The reviewers have commented on your above paper. They indicated that it is not acceptable for publication in its present form.

However, if you feel that you can suitably address the reviewers' comments, I invite you to revise and resubmit your manuscript.

Reviewer 1 ·

Basic reporting

.

Experimental design

.

Validity of the findings

.

Additional comments

The manuscript describes the potential of 3D tracking of zebrafish locomotion for the analyses of two compounds. Several major and minor issues should be addressed:

Major issues:

1. Public availability of the 3D tracking sotware used is indispensable for an adequate evaluation of the manuscript, in terms of scientific transparency and possibilities of reproducibility of their findings. The authors state in this manuscript: "Images recorded from the high-resolution cameras were processed through an in-house developed tracking software, see (Butail et al. 2013) for a detailed description". A close inspection of the Butail et al, 2013 paper did not find that the 3D tracking sotware used is publicly available. Other papers of the same group mention it but does not include the public availability of the 3D tracking sotware used: Butail S, et al, PLOS ONE 2013 (10.1371/journal.pone.0076123); Ladu F, et al, Zebrafish. 2015 (10.1089/zeb.2014.1041); Macri S, et al, Sci Rep. 2017 (10.1038/s41598-017-01990-z). Without public availability of a working version of the 3D tracking sotware used, it is impossible for this reviewer and for other scientists to verify or replicate their findings.
2. An analysis of posible false positives and false negatives should have a different statistical approach.

Minor points:
-The titie should be modified, in order to have a better reflection of the actual findings of the study.
-"genetic and neuroanatomic isomorphism" should be revised
-"High-throughput behavioral experiments on zebrafish generally share the following methodological structure: administration of water-soluble drugs" should be revised, as it also involves other type of stimuli.
-"we aimed at prospectively investigating whether 3D scoring of zebrafish behavior may also benefit pharmacological research", should be revised, as alcohol is more related to toxicology.
-Authors should explain the reasons for use of "serotonin reuptake inhibitor citalopram". Citalopram is mainly used as an antidepressant in humans.
-Details of preliminary PCA analyses are better located in the Methods section.
-Authors should provide details of the "stress coat" used
-Authors should provide details of the Flea 3 high resolution cameras and associated videos
-Authors should highlight that some tanks for traditional 2D analyses restrict movement in one of the axes
-Authors should highlight that the "3D analysis" is also biased to restrict movements in one of the axes (width is only 8.5 cm)
-Authors should include details of statistical significance in figures.
-Authors should discuss the posible disadvantages or challenges of the proposed 3D tracking analysis

Reviewer 2 ·

Basic reporting

-The title should be changed in order to describe clearly the study and its results
-The aims of the study should be described as clearer and more concise.
-Authors should explain why they used citalopram and ethanol
-The last part of the introduction, about PCA is redundant, this part is in statistical and in the results section.
-The legends of the figures should have more information for a better interpretation, for example, the statistical method used, P-value. In figure 1 is important to mention what software was used to the reconstruction of the trajectory.

Experimental design

-Authors should report of resolution of the cameras
-what was the distance between the camera in overhead and the bottom of the tank ?, and -what was the distance between the camera in the front and the bottom of the tank?
-The tracking software is available for anyone?

Validity of the findings

-In 3.1 section subtitle, authors should clarify that PCA was performed only for 3D analysis
-Authors should conclude and discuss clearly if the model was or not replicated, according to their proposed goals in the introduction. For example, in the introduction, they mentioned that one of the goals of this study was” to replicate existing findings indicating that ethanol (Pannia et al. 2014)”. However, in the discussion section, the authors no compared their results or discuss it with Pannia´s results.
-It is not clear how the authors determined the false positive and false negative, this should be described in the method section

---

## Round 0.2 · accepted · Accept

Thank you very much for improving your manuscript.

Reviewer 1 ·

Basic reporting

No comment

Experimental design

No comment

Validity of the findings

No comment

Additional comments

The authors incorporated the suggestions made by this reviewer. The revised manuscrip might be accepted for publication.

Reviewer 2 ·

Basic reporting

The authors made the suggested corrections

Experimental design

The authors made the suggested corrections

Validity of the findings

The authors made the suggested corrections